# *NR4A2* as a Novel Target Gene for Developmental and Epileptic Encephalopathy: A Systematic Review of Related Disorders and Therapeutic Strategies

**DOI:** 10.3390/ijms25105198

**Published:** 2024-05-10

**Authors:** Alba Gabaldon-Albero, Sonia Mayo, Francisco Martinez

**Affiliations:** 1Translational Research Group in Genetics, Instituto de Investigación Sanitaria La Fe, 46026 Valencia, Spain; alba_gabaldon@iislafe.es; 2Genetics and Inheritance Research Group, Instituto de Investigación Sanitaria Hospital 12 de Octubre, 28041 Madrid, Spain; 3Department of Genetics, Hospital Universitario 12 de Octubre, 28041 Madrid, Spain; 4Genetics Unit, Hospital Universitario y Politecnico La Fe, 46026 Valencia, Spain

**Keywords:** *NR4A2*, *NURR1*, intellectual disability, language impairment, epilepsy, parkinsonism, dystonia, Alzheimer’s disease

## Abstract

The *NR4A2* gene encodes an orphan transcription factor of the steroid–thyroid hormone–retinoid receptor superfamily. This review focuses on the clinical findings associated with the pathogenic variants so far reported, including three unreported cases. Also, its role in neurodegenerative diseases, such as Parkinson’s or Alzheimer’s disease, is examined, as well as a brief exploration on recent proposals to develop novel therapies for these neurological diseases based on small molecules that could modulate *NR4A2* transcriptional activity. The main characteristic shared by all patients is mild to severe developmental delay/intellectual disability. Moderate to severe disorder of the expressive and receptive language is present in at least 42%, while neuro-psychiatric issues were reported in 53% of patients. Movement disorders, including dystonia, chorea or ataxia, are described in 37% patients, although probably underestimated because of its frequent onset in late adolescence–young adulthood. Finally, epilepsy was surprisingly present in 42% of patients, being drug-resistant in three of them. The age at onset varied widely, from five months to twenty-six years, as did the classification of epilepsy, which ranged from focal epilepsy to infantile spasms or Lennox–Gastaut syndrome. Accordingly, we propose that *NR4A2* should be considered as a first-tier target gene for the genetic diagnosis of developmental and epileptic encephalopathy.

## 1. Introduction

*NR4A2* gene (MIM *601828), also known as *NURR1*, encodes for nuclear receptor subfamily 4 group A member 2, a member of the steroid–thyroid hormone–retinoid receptor superfamily. The encoded protein is an orphan transcription factor, which acts through binding to specific genomic DNA response element sequences. The protein contains two prominent domains: (1) a zinc finger C4 type domain (positions 260-335) that acts as DNA-binding domain; and (2) a ligand-binding domain (positions 360-595), also called HOLI domain (UniProt P43354-1). Until recently, it was considered that NR4A family members were orphan nuclear receptors, with no physical space in this domain as a result of the tight packing of side chains from several bulky hydrophobic residues in the region normally occupied by ligands [1], which is what suggested that these receptors may function independent of binding ligand within an orthosteric pocket [2]. However, recent nuclear magnetic resonance studies of the isolated HOLI domain suggested that several synthetic small molecules could bind in canonical and non-canonical binding pockets [3] or to allow binding of unsaturated fatty acids [4]. Although still unknown, these putative endogenous ligands might constitute regulatory signals with the capacity to modify the transcriptional activity through conformational changes [5,6].

NR4A2, expressed in embryogenesis, is essential not only for early differentiation of midbrain dopaminergic neurons [7], but it is also required for maintenance of these adult neurons [8]. Widely expressed in the central nervous system, this transcriptional regulator is crucial for the expression of a set of genes such as *SLC6A3*, *SLC18A2*, *TH*, and *DRD2* which are essential for the differentiation and maintenance of meso-diencephalic dopaminergic neurons during development [2,9]. In addition, Saijo et al. [10] found that a reduced *NR4A2* expression resulted in exaggerated inflammatory responses in microglia that were further amplified by astrocytes, leading to production of factors that caused the death of tyrosine hydroxylase-expressing neurons. Alternatively, spliced isoforms have been described, albeit currently unknown whether these different isoforms might have any specific role.

Mutations in this gene have been reported in patients with intellectual developmental disorder with language impairment and early-onset DOPA-responsive dystonia-parkinsonism (IDLDP; MIM #619911) and have been associated with disorders related to dopaminergic dysfunction, including Parkinson’s disease (PD), schizophrenia, and manic depression [11,12]. In addition, some authors have suggested that a misregulation of this gene might also be associated with autoimmune inflammatory diseases such as rheumatoid arthritis, psoriasis, or multiple sclerosis [13,14,15]. IDLDP is an autosomal dominant disorder characterized by global developmental delay affecting motor, cognitive, and speech domains apparent in early childhood or infancy. Most patients also show movement abnormalities, often hypotonia with later development of dopa-responsive dystonia or parkinsonism, and about half of the patients develop various types of seizures [16,17,18,19].

This systematic review in *NR4A2* is focused on three major points: (1) collecting the clinical findings, including three unreported cases, and actualizing the phenotype associated to pathogenic variants in *NR4A2*; (2) examining its role in neurodegenerative diseases such as Parkinson’s or Alzheimer’s disease; and (3) exploring target-based therapies for these neurological diseases.

## 2. Methods

Electronic research was conducted to identify pertinent English articles examining the genetic associations between *NR4A2* and abnormal nervous system physiology (HP:0012638), following the PRISMA guidelines [20]. Two types of databases were consulted: PubMed for publications and ClinVar (at URL https://www.ncbi.nlm.nih.gov/clinvar/ (accessed on 20 March 2024)) and the Human Gene Mutation Database (HGMD) Professional for specific case reports (Figure 1). Therefore, studies with both clinical and preclinical information were included for this review.

A systematic literature research of PubMed was performed to identify eligible articles until 20 March 2024, using the following search terms: (NR4A2 OR NURR1) AND (Neurodevelop* OR intellect* OR Language OR speech OR Intellectual OR epilep* OR seizur*). The search identified 78 potential articles.

Reviews were excluded. No systematic reviews, editorials, commentaries, or articles in a different language than English were reported until that date. Titles and abstracts were screened to check if they fell within the scope of this review. In some cases, when more information was required to decide, a quick review of the whole article was carried out. At this stage, a total of 31 original articles were selected.

For clinical cases, ClinVar and HGMD Professional databases were revised. From the 43 pathogenic variants for *NR4A2* reported in ClinVar, 22 were Copy Number Variants (CNVs) affecting more genes (12 deletions and 14 duplications). One variant was excluded because no condition associated to it was provided. From the other 20 variants, 6 were described by Jesús et al. (2021), Singh et al. (2020), or Lévy et al. (2018) and are included in this systematic review [18,19,21]. From the remaining 134 variants, only one was reported in a study not included in the PubMed search [22].

In relation to HGMD Professional, 30 deleterious variants were reported, and only 2 deletions were excluded for affecting more genes and 1 was excluded because it was associated to Cystic kidney disease from a source not available and verifiable. From the other 27 variants, 15 were described in 9 different publications and are included in this systematic review [16,17,18,19,21,23,24,25,26]. Another nine variants were reported in nine studies not included in the PubMed search [11,12,27,28,29,30,31,32,33] and three variants were reported by multiple submitters, including four not included in this systematic review [34,35,36,37].

## 3. Results

### 3.1. Case Reports—Clinical Findings

The phenotypes associated with the 30 pathogenic or likely pathogenic reported variants include: psychiatric disorders (schizophrenia and bipolar disorder), late onset Parkinson’s disease, parkinsonism and dopa-sensitive dystonia, and intellectual developmental disorder. Phenotypic variability can be noted since more than one of these phenotypes have been described for the same variant (Figure 2).

Detailed descriptions of 23 patients with pathogenic variants in *NR4A2* are available in the literature [16,17,18,19,21,23,25,26,30,38,39], of which 7 had a loss of a copy of the gene because of deletions of variable size. The latter cases are not included in the text due to the potential impact on the phenotype of the neighboring deleted genes; however, their descriptions are summarized in Table 1. On the other hand, we have included three patients with pathogenic variants not previously described [40].

The three novel patients share a common history of an uneventful pregnancy and neonatal period, followed by significant communication delay, especially affecting expressive language development and mild-moderate intellectual disability with psychiatric issues. Patient 17, at eleven years old, expresses herself in sentences of up to three words, understands simple commands, and has not yet acquired literacy. Major concerns now are related to attention-deficit/hyperactivity disorder (ADHD), aggressive behavior and sleep disorder. In the motor sphere she has difficulties with buttoning and tying and has recently learned to swim. Patient 18 is now twenty-five years old. Her first words were at the age of three years; she achieved literacy but has difficulties in reading comprehension. She is currently independent in daily activities but has significant difficulties in expressive language and fine motor skills, such as writing. She presents behavioral disorder with aggression and delusional thoughts, which began after the debut of focal epilepsy at the age of 22. After starting anti-dopaminergic treatment, she presented oculogyric crisis. Her epilepsy has been refractory to several lines of anti-seizure medications. Patient 19 is now thirty-two years old. Her first words were at the age of three years and a half. She reached literacy with difficulty, needing curricular adaptation during primary education. Currently she has difficulties in expressing herself. She is temperamental and has a low tolerance to frustration together with aggressive behavior. She is autonomous in daily life. Epilepsy started when she was twenty-six years old, being focal and drug resistant. MRI showed small foci of polymicrogyria in both frontal opercula.

Therefore, we summarize here the clinical characteristics based on a total of 19 cases (Table 1). At the last clinical assessment, the patients ranged in age from 2 to 32 years. No abnormalities in prenatal or perinatal history were detected in any case; growth parameters were normal. Developmental delay and/or intellectual disability was the characteristic shared by all the patients, ranging from mild to severe. Language disorder was the second most frequent, being reported in 14 patients. In one case, expressive language was absent at 6 years [39], while in eight patients, a moderate to severe disorder of the expressive and receptive language was noted; information on this issue was unavailable in five cases. Learning disabilities were reported in six cases, most frequently related to the acquisition of reading and writing. Neuropsychiatric issues were described in 10 patients and included aggressive behavior, anxiety, ADHD, and autism.

Regarding motor development, it was delayed in nine cases, and hypotonia was reported. However, none of them were reported to not achieve independent walking, although in one case, it only was achieved after treatment with L-dopa [25].

Movement disorder was described in seven patients. In only one of them did the symptoms appear in pediatric age, with the most frequent onset between late adolescence and young adulthood. The phenotypes described were adult-onset dystonia-parkinsonism (3/19), isolated dystonia (2/19), dystonia combined with chorea (1/19), ataxia with motor tics (1/19), and isolated ataxia (1/19). It is worth mentioning the evolutionary character of the semiology described in some of these cases, which may start as tics, progresses to focal dystonia and then to dystonia-parkinsonism. In one case, focal dystonia in the form of oculogyric crises started in the context of treatment with olanzapine, a dopaminergic antagonist, which could be due to an increased vulnerability to extrapyramidal symptoms with the use of these drugs.

Epilepsy was described in eight cases, being drug-resistant in three. The age at onset was variable, ranging from the five months to twenty-six years of age. The seizures were described as focal motor with impaired awareness (6/19), epileptic spasms (2/19), generalized tonic–clonic (1/19), generalized tonic and myoclonic (1/19), or absence and atonic (1/19). Epilepsy could be classified as focal epilepsy (5/19), infantile spasms (2/19), Lennox–Gastaut syndrome (1/19), generalized (1/19), or generalized and focal epilepsy (1/19).

The interictal electroencephalogram describes hypsarrhythmia in the cases of infantile spasms and sharp waves, spikes, or spike-waves especially in temporal regions, sometimes affecting central or parietal areas. In one case, epileptiform activity and photosensitivity without seizures was reported.

Finally, no characteristic structural signs were detected by magnetic resonance imaging (MRI) of the brain, although anomalies were described in five cases: basal ganglia and thalamic gliosis, cerebellar atrophy, pontine hypoplasia, ventriculomegaly, operculo-insular polymicrogyria, and enlarged cerebrospinal fluid (CSF) spaces. In functional neuroimaging studies (123I-FP-CIT SPECT) performed in cases with movement disorder, reduced uptake was observed at the level of the striatum, along with biochemical evidence in CSF of dopaminergic dysfunction (reduced homovanillic acid/5-hydroxyindoleacetic ratio).

### 3.2. NR4A2 and Neurodegenerative Disease

Given the functions of *NR4A2* in the maintenance and survival of dopaminergic neurons in the midbrain [41], as well as in the hippocampus and cortex [42], a dysregulation of its activity was proposed to play a role in the onset or progression of neurodegenerative diseases such as Parkinson’s or Alzheimer’s disease.

In relation with Parkinson’s disease, conditional knockout mice were shown to mimic the early aspects of Parkinson’s, demonstrating locomotor deficits and a series of pathological alterations such as the loss of dopaminergic neurons and decreased striatal dopaminergic neurons [43]. Even heterozygous conditional knockout (KO) mice for *Nr4a2* developed a progressive loss of dopaminergic neurons [8]. Furthermore, Nr4a2 protein agonists have been shown to reverse behavioral and histological abnormalities in animal models of Parkinson’s [3]. It has been proposed that NR4A2 protects dopaminergic neurons from inflammation-induced degeneration, and more specifically attenuates neuronal death by suppressing the expression of inflammatory genes in microglia and astrocytes (reviewed in [44]).

These kinds of results prompted the search for genetic variants that cause or facilitate this prevalent degenerative disease that affects 1–2% of the population over the age of 65. Initially, two genetic variants in the untranslated region of exon 1 of the *NR4A2* gene in humans were proposed to cause familial Parkinson’s disease (PD), being associated to a decrease mRNA expression in transfected cell lines and in lymphocytes of affected individuals [12]. However, subsequent studies failed to replicate these results, concluding that sequence alterations in exon 1 of the *NR4A2* gene are not a major cause of familial Parkinson’s [45,46,47,48]. Thus, the aforementioned variants were reclassified in OMIM (MIM *601828) as variants of unknown significance. On the other hand, Liu et al. (2017) performed a systematic review of a total of 34 published studies on the presumed association of *NR4A2* with PD. After meta-analysis, no significant association was observed, except for a very modest association with a highly frequent intronic variant in the *NR4A2* gene (c.1361 + 18delG; rs35479735) assuming a recessive and homozygous model (OR of 1.31, *p* = 0.003) [49]. In conclusion, the contribution of genetic variants in this gene to PD can be considered very scarce.

In relation to Alzheimer’s disease (AD), it was found that expression of a dominant-negative variant of Nr4a in the mouse forebrain, including the hippocampus, impaired long-term hippocampus-dependent contextual fear memory without affecting short-term contextual fear memory or hippocampus-independent cued fear memory [50]. A great deal of evidence has highlighted the role of Nr4a2 in hippocampus-dependent cognitive functions (reviewed in [51]). Indeed, a specific decrease in Nr4a2 mRNA levels were found in Aβ1–42-treated neuronal cells [52] and in mouse models. A similar decrease was also reported in postmortem brains of human AD patients, specifically in the frontal cortex and hippocampal formation [53,54]. Additional support comes from the observation that the number of Nr4a2-expressing cells significantly decreased in a mouse model in an age-dependent manner, accompanied by increased plaque deposition, suggesting a possible cause–effect relationship between the Nr4a2 levels and AD progression [42]. Most interestingly, Moon et al. found that knockdown of Nr4a2 significantly aggravated AD pathology while its overexpression alleviated it, including a decrease in Aβ accumulation and neurodegeneration [54].

### 3.3. Treatments

Even in absence of genetic variants in the *NR4A2* gene directly implied as the cause of Parkinson’s or Alzheimer’s disease, there is strong evidence for the importance of a putative pharmacological modulation of this gene to revert or ameliorate the symptoms in these highly prevalent diseases. As stated above, NR4A2 protein agonists have been shown to reverse behavioral and histological abnormalities in animal models of Parkinson’s [3]. On the other hand, the benefit that this treatment would mean for patients with the rare NR4A2-related developmental disorder is unquestionable. So far, a targeted treatment is limited by the absence of proven, effective chemical tools. However, the identification of synthetic small molecules that could modulate NR4A2 transcriptional activity opens the possibility to provide target-based therapies for these neurological diseases. As mentioned above, some findings suggest that NR4A2 binds fatty acid metabolites and fatty acid mimetic (FAM) drugs, opening new opportunities for NR4A modulator development. Very recently, new FAM agonists and inverse agonists with sub-micromolar potency and binding affinity demonstrated remarkable potential as NR4A-modulating tools and drugs [55]. In preclinical studies, these compounds significantly improved the behavioral deficits in rat models of PD without any noticeable sign of dyskinesia-like behavior [3]. Autophagic-lysosomal blockade was also reported as one of their functions [56]. Recently, the antimalarial drug amodiaquine has also been found to enhance cognitive functions by increasing adult hippocampal neurogenesis [57]. More recently, a mouse model of AD treated with this agonist showed robust reduction in typical AD features including deposition of amyloid beta plaques, neuronal loss, microgliosis, and impairment of adult hippocampal neurogenesis, leading to significant improvement in cognitive functions [54].

On the other hand, clinically relevant neuroprotective effects have been demonstrated for statins. After screening a drug fragment library for NR4A2 modulation in a cellular setting, statins demonstrated to be potent NR4A2 modulators [58]. Several statins directly improve its transcriptional activity both in cellular and cell-free conditions with low-micromolar to sub-micromolar potencies. Simvastatin treatment, for example, was shown to be capable of inducing anti-inflammatory effects in astrocytes, as well as other neuroprotective changes including improved glucose metabolism, energy generation or the expression of genes related with cell cycle. These neuroprotective changes are abrogated by *NR4A2* knockdown [58].

Also, the clinically studied dihydroorotate dehydrogenase (DHODH) inhibitor vidofludimus calcium has been proposed as NR4A2 agonist with strong activation efficacy at nanomolar potency. The optimized compound induced NR4A2-regulated gene expression in astrocytes and exhibited favorable pharmacokinetics in rats, thus emerging as a chemical tool to study NR4A2 activation in vitro and in vivo [59].

Other alternative approaches have been proposed to regulate the activity of NR4A2, including post-translational modifications, among which phosphorylation by kinases or SUMOylation are the best characterized (reviewed in [60]).

Finally, a novel mode of activation of this nuclear receptor has been recently proposed, based on the use of ligands that target its heterodimer-binding partner, retinoid X receptor alpha (RXRα; NR2B1), which in turn promotes the heterodimer dissociation, enhancing the transcriptional activity of NR4A2 on monomeric DNA response elements [61].

## 4. Discussion

We have seen *NR4A2* gene codes for a transcription factor implied in neuronal differentiation and maintenance not only of dopaminergic neurons, but also in the hippocampus and in other areas of the forebrain. It is therefore not surprising that the pathogenic variants in this gene not only cause movement disorders, mainly dystonia, but also a complex neurodevelopmental disorder with special affectation of language and a high predisposition to neuropsychiatric disorder. There is a wide phenotypic heterogeneity, and the severity is highly variable, in such way that the less severely affected individuals have only mild deficits and are able to attend schools for individuals without disabilities. Conversely, evidence for the implication of pathogenic variants in Parkinson’s or Alzheimer’s diseases are very scarce, despite being a good candidate target for therapies designed to ameliorate or even reverse the symptoms of theses neurodegenerative diseases of high prevalence.

Among the reported pathogenic variants in the *NR4A2* gene, there were nine missense, six frameshift, two nonsense, and two splicing variants (see Table 1). It is worth noting that all of them were de novo, except for two cases in which this information was not available. This is in accordance with the autosomal dominant pattern of inheritance, which is by far the most frequent mode of inheritance in all neurodevelopmental disorders and probably in the known neurological diseases of Mendelian cause.

All the patients with pathogenic or likely pathogenic variants in *NR4A2* show developmetal delay and/or intellectual disability. Movement disorders are usually of late onset, being reported in 37% (7/19) of the patients in the whole series, but this percentage increases to 55% (5/9) among patients over 18 years, and 80% (4/5) among patients over 30 years old. Epilepsy is also a frequent sign, reported in 42% (8/19) patients. There is no clear correlation between the type of mutation (missense, truncating) or its location along the gene with epilepsy. Although epilepsy seems to be more frequent among patients with truncating mutations (6/10) than with missense variants (2/9), this difference does not achieve statistical significance (*p* = 0.168; Fisher exact test). The missense variants in the two patients with epilepsy are located in the DBD and the LBD domains, respectively.

Recent studies support the role of *NR4A2* in hippocampal synaptic plasticity and cognitive functions, although the underlying molecular mechanisms are still poorly understood (revised in [51,60]). Psychiatric, communicative, and cognitive problems may be related to the underlying dysfunction of dopaminergic pathways comprising the mesolimbic system, as these are involved in mood and attention regulation, working memory, and associative learning [60,62]. In addition, since primary language areas receive dopaminergic projections from subcortical regions [63], dysfunction of these pathways may influence the regulation of language processing. Furthermore, dopamine also plays a role in the executive functions via the mesocortical pathway, which are necessary for the organization of speech in expressive language and proper modulation of attention, necessary for adequate auditory processing and language comprehension [64].

Even less is understood about how the pathogenic variants of *NR4A2* can contribute to epileptogenesis (Figure 3). *NR4A2* contains a bipartite nuclear localization signal (NLS) within its DBD domain and three leucine-rich nuclear export signals (NES) in its LBD. Together, these signals regulate NR4A2 shuttling in and out of the nucleus, and it has been detected in the postsynaptic density fraction of synaptoneurosomes obtained from adult mouse hippocampi so that NR4A2 could also be considered as being localized at synapsis [64]. On the other hand, *NR4A2* contains a “half-CRE” site in its promoter, and in fact, *NR4A2* is a downstream target of CREB activation in many systems, including the hippocampus [65,66]. In turn, the NR4A2 activation is responsible for the increase in *BDNF* expression linked to NMDA prosurvival effects [66]. This brain-derived neurotrophic factor (BDNF) participates in axonal growth, pathfinding, and the modulation of dendritic growth and morphology, being a major regulator of synaptic transmission and plasticity at adult synapses in many regions of the central nervous system, so that it has been tentatively implicated in the pathogenesis of epilepsy [67].

## 5. Conclusions

Pathogenic variants in *NR4A2* gene cause a complex neurodevelopmental disorder with special affectation of language, as well as dystonia or other movement disorders, neuropsychiatric disorder and/or epilepsy. Almost half of the patients present epilepsy, which may initiate at any age and in a diverse semiology, accompanied by the aforementioned complex and in many occasions severe neurodevelopmental disorder. Also, it is important to note that some patients may have normal early development in infancy before symptom onset. According to these findings, we propose that *NR4A2* should be considered as a first-tier candidate gene for study in patients with suspected developmental and epileptic encephalopathy. Although pathogenic variants in this gene have been rarely reported so far, the high variability of the clinical presentation lets us suppose that it is, in fact, underdiagnosed. Most importantly, there are high expectations of achieving a treatment targeted at this gene.

## Figures and Tables

**Figure 1 ijms-25-05198-f001:**
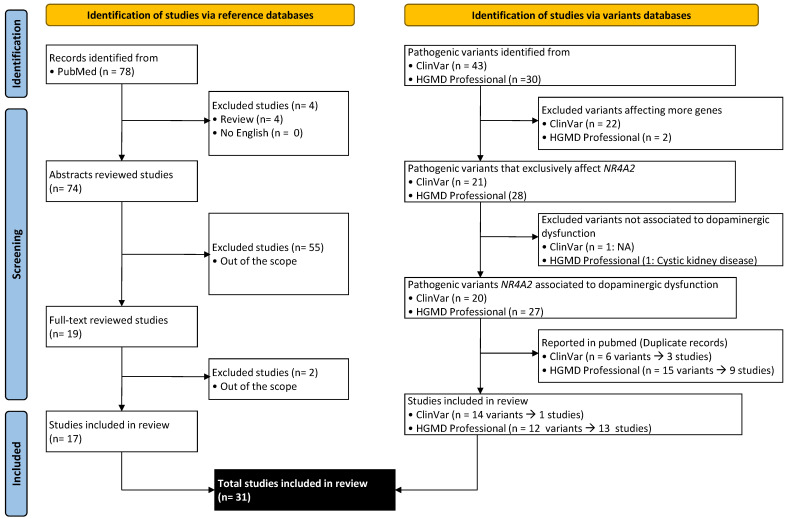
Flow diagram summarizing the identification, screening, and studies selection from PubMed, ClinVar, and HGMD Professional based on a PRISMA 2020 flow diagram template [20].

**Figure 2 ijms-25-05198-f002:**
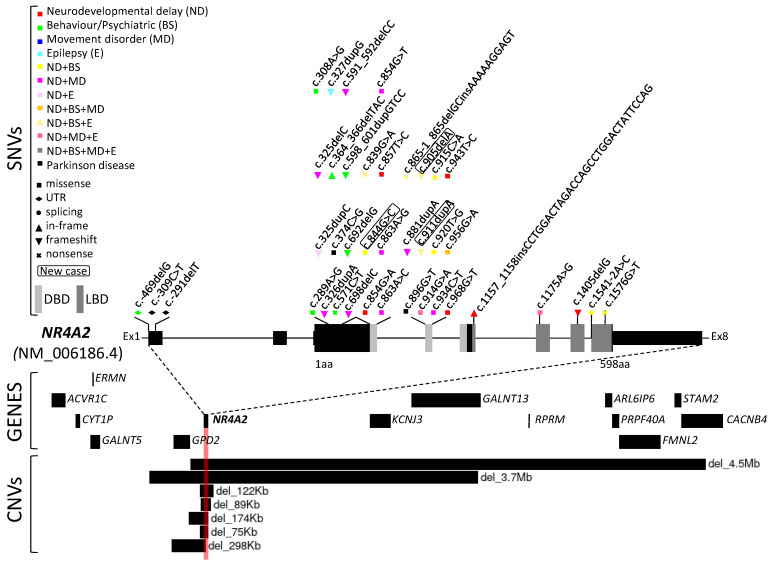
Schematic representation of the reported pathogenic/likely pathogenic variants and their main phenotype in *NR4A2.* DBD: DNA-binding domain; LBD: ligand-binding domain.

**Figure 3 ijms-25-05198-f003:**
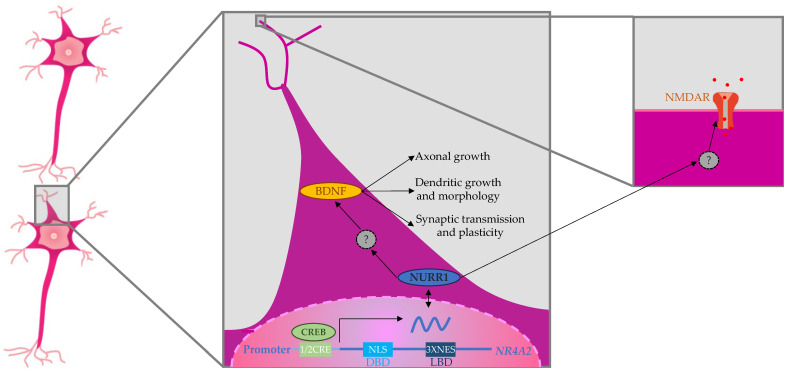
Proposed mechanism to explain the role of *NR4A2* in synaptic plasticity, cognitive functions, and epilepsy.

**Table 1 ijms-25-05198-t001:** Clinical and genetic features.

**Reference**	**Patient 1** [30]	**Patient 2** [16]	**Patient 3** [17]	**Patient 4** [17]	**Patient 5** [18]	**Patient 6** [18]	**Patient 7** [18]	**Patient 8** [18]	**Patient 9** [18]
Age (years)/Sex	15/F	9/M	32/M	57/F	15/F	12/M	9/F	3/F	5/M
Motor milestones	Delayed	Normal	Normal	Clumsiness	Delayed	Normal	NA	Delayed	Delayed
Hypotonia	NA	No	No	NA	NA	Yes	NA	Yes	Yes
ID severity	NA	Mild	Mild	Mild	Severe	Mild	Mild-moderate	Severe	Mild
Language	NA	Delayed	Delayed	Normal	NA	LD	NA	NA	LD
Psychiatric and behavioral	NA	No	No	No	ASD	Hyperactivity Anxiety	NA	No	Attachment disorder, ADHD
Epilepsy/Age at onset	No	Yes/5 years	Yes/26 years	No	Yes/6.5 years	Yes/10 years	No	Yes/5 months	No
Seizure type/Epilepsy classification	-	Rolandic epilepsy	Generalized tonic-clonic seizures	-	Focal motor	Rolandic epilepsy	-	Infantile spasms	-
Refractory epilepsy	No	No	NA	-	No	No	-	No	-
Movement disorder/Age at onset	-	No	Dystonia- Parkinsonism/ 29 years	Dystonia- Parkinsonism/ 30 years	No	No	Dystonia Choreo-athetosis Ataxic gait/NA	Dystonia/ NA	No
Dopa responsive	NA	-	Yes	Yes	-	-	NA	NA	-
Other signs	-	-	-	-	-	RWLD	-	Microcephaly CVI	Hyposensitivity CVI
MRI	NA	Normal	NA	Thinning of substantia nigra	Normal	Normal	Gliosis of thalamus and basal ganglia.	Cerebellar atrophy	NA
Variant (NM_006186.3)	c.920T > G	c.326dupA	c.326dupA	c.881dupA	c.839G > A	c.865-1_865delGCinsAAA	c.914G > A	c.1175A > G	c.1576G > T
Type	Missense	Frameshift	Frameshift	Frameshift	Missense	Splicing	Missense	Missense	Nonsense
de novo	Yes	Yes	Yes	NA	Yes	Yes	Yes	Yes	NA
**Reference**	**Patient 10 [18]**	**Patient 11 [18]**	**Patient 12 [18]**	**Patient 13 [19]**	**Patient 14 [25]**	**Patient 15 [23]**	**Patient 16 [23]**	**Patient 17**	**Patient 18**
Age (years)/Sex	2/M	4/F	19/F	30/M	2.5/M	11/M	12/M	11/F	25/F
Motor milestones	Delayed	Delayed	Delayed	Clumsiness	Delayed	Normal	Delayed	Clumsiness	Normal
Hypotonia	Yes	Yes	Yes	No	Yes	Yes	Yes	No	No
ID severity	Severe	Moderate	Severe	Mild	Mild-moderate	Mild	Mild	Mild	Moderate
Language	Normal	LD	LD	LD	Delay	LD	Delay	LD	LD
Psychiatric and behavioral problems	No	No	No	ADHD Trichotilomania	No	ADHD	ADHD	ADHD	Delusions, hallucinations, and mood swings
Epilepsy/Age at onset	Yes/6 months	No	No	No	NA	No	No	No	Yes/23 years
Seizure type/Epilepsy classification	Infantile spasms Lennox–Gastaut Syndrome	-	-	-	-	-	-	-	Focal (motor)
Refractory epilepsy	Yes	-	-	-	-	-	-	-	Yes
Movement disorder/Age at onset	No	No	No	Motor tics, cervical dystonia, dystonia-parkinsonism/16 years	Multifocal dystonia/22 months	No	No	No	Oculogyric crisis/ 24 years
Dopa responsive	-	-	-	NA	Yes	-	-	-	NA
Other signs/symptoms	Sensory and sleep disorder	-	-	-	-	RWLD	RWLD	RWLD	RWLD
MRI	Pontine hypoplasia Ventriculomegaly	Normal	Normal	Normal	Normal	Normal	Normal	Normal	Ventriculomegaly Widened subarachnoid space
Variant (NM_006186.3)	c.325dupC	c.857T > C	c.968G > T	c.956G > A	c.863A > G	c.1541-2A > C	c.915C > A	c.844G > C	c.905delA
Type	Frameshift	Missense	Missense	Missense	Missense	Splicing	Nonsense	Missense	Frameshift
de novo	Yes	Yes	Yes	Yes	Yes	Yes	Yes	Yes	Yes
**Reference**	**Patient 19**	**Patient 20 [38]**	**Patient 21 [26]**	**Patient 22 [21]**	**Patient 23 [21]**	**Patient 24 [21]**	**Patient 25 [39]**	**Patient 26 [18]**	
Age (years)/Sex	32/F	25/F	8/F	17/M	8/F	9/M	6/F	43/M	
Motor milestones	Clumsiness	Normal	Delayed	Normal	Normal	Normal	Delayed	Delayed	
Hypotonia	No								
ID severity	Mild	Moderate	Mild	Mild-moderate	NA	NA	Severe	Severe	
Language	LD	Delay	LD	Delay	LD	LD	LD	NA	
Psychiatric and behavioral problems	Behavioral disorder	NA	No	ASD	Restlessness	ASD	NA	ADHD	
				ADHD		Behavioral disorder		Behavioral disorder	
Epilepsy/Age at onset	Yes/26 years	NA	NA	No	No	No	NA	Yes/13 years	
Seizure type/Epilepsy classification	Focal (motor)	-	-	-	-	-	-	Lennox–Gastaut Syndrome	
Refractory epilepsy	Yes	-	-	-	-	-	-	Yes	
Movement disorder/Age at onset	No	No	No	No	No	No	NA	Ataxia/adulthood	
Dopa responsive	-	-	-	-	-	-	-	NA	
Other signs/symptoms	RWLD	-	-	RWLD	-	-	-	-	
MRI	Bilateral operculo-insular polymicrogyria foci	Normal	NA	NA	NA	NA	Normal	Widened subarachnoid space	
Variant (NM_006186.3)	c.911dupA	2q24.1(157161283-157459740)×1	2q24.1(157120975–157210126)×1	2q24.1(157094848_157216692)×1	2q24.1(157141281_157216692)×1	2q24.1(157141280_157315748)×1	2q23.3q24.1(152796289–157299545)×1	2q23.3q24.1(154790212_158488241)×1	
Type/size	Frameshift	298 kb	89 kb	122 kb	75 kb	174 kb	4.5 Mb	>3.6 Mb	
de novo	Yes	Yes	Yes	Yes	Yes	Yes	Yes	Yes	

F Female, M Male, NA Not Available, ID Intellectual Disability, LD Language Disorder, ASD Autism Spectrum Disorder, IA Impaired Awareness, CVI Cerebral Visual Impairment, RWLD Reading and Writing Learning Disorder, cMR Cerebral Magnetic Resonance, cCT Cerebral Computed Tomography.

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
