# Peer review of "NR4A2 as a Novel Target Gene for Developmental and Epileptic Encephalopathy: A Systematic Review of Related Disorders and Therapeutic Strategies"

_ijms, 2024, doi:10.3390/ijms25105198_

Round 1

Reviewer 1 Report

Comments and Suggestions for Authors

Alba Gabaldon-Albero's manuscript, "NR4A2 as a novel target gene for developmental and epileptic encephalopathy: a systematic review of related disorders and therapeutic strategies," is a systematic review of the current status of NR4A2 in neurological diseases and suggests that the NR4A2 gene can be used as a targeted gene for the diagnosis of developmental and epileptic encephalopathy.  This review may be considered for publication with major revisions.

 Major Comments:  

 1. In the discussion, authors should address the mechanism of NR4A2 in relation to developmental and epileptic encephalopathy.

2. Authors must include a pathway diagram or graphical depiction to clearly understand how NR4A2 may play a role in developmental and epileptic encephalopathy.

3. Authors must highlight any limitations or cautions when investigating the NR4A2 targeted gene for developmental and epileptic encephalopathy.

Minor Comments:

Please check for a few typos.

Line 16- This review focuses on

Line 23 - dystonia is repeated twice

Line 61- intellectual I should not be capitalized as it is not a proper noun

Line 71- about half of the patients

Line 91 - could be rewritten as 'Titles and abstracts were screened to check if they fell within the scope of this review.' 

Line 140 - font is different for the in text citation 

Line 323- ....are very scarce, despite being a good candidate target...

Author Response

We appreciate the comments and suggestions kindly proposed by the reviewer, which we address as follows:

  1. In the discussion, authors should address the mechanism of NR4A2 in relation to developmental and epileptic encephalopathy.

A discussion about the putative mechanisms has been included (see lines 360 to 384).

  1. Authors must include a pathway diagram or graphical depiction to clearly understand how NR4A2 may play a role in developmental and epileptic encephalopathy.

Figure 3 has been developed in order to try to illustrate the proposed mechanism to explain NR4A2 role in synaptic plasticity, cognitive functions and epilepsy

  1. Authors must highlight any limitations or cautions when investigating the NR4A2 targeted gene for developmental and epileptic encephalopathy.

We are not quite sure about what is required. Genetic studies aimed at diagnosis are currently not carried out gene by gene, but in physical gene panels (targeted NGS of a certain group of genes by hybridization capture or by amplification) or virtual panels, when the genes of interest or target genes are selected for the analysis of variants in a whole exome or genome study. We understand that the limitations or precautions of including the NR4A2 gene in the target genes when studying a case with suspicion of developmental and epileptic encephalopathy are the usual in this type of study, but with the opportunity to increase the possibility of finding the causal genetic variant, and therefore the sensitivity of the study.

Minor Comments:

The few typos have been amended, thank you.

Reviewer 2 Report

Comments and Suggestions for Authors

This review has an interesting topic. In recent years, especially in the field of neuroscience and in the management of intractable disease, such as developmental and epileptic encephalopathy (DEE), precision medicine plays a significant role. Knowledge of the genetic substrate, the genes involved and their mutation, allows to set up a personalized therapy for each patient. An example is the spectrum of SCN1A epileptic disorders other than Dravet syndrome (DS). Find new variants of the gene allows to optimize the  antiseizure management, correlating phenotype-genotype.

This review is well conducted, but minor improvements are needed.

- "Results and discussion" paragraphs could be separated to make the manuscript clearer. In particular, the section, that explore the correlation between NR4A2 and neurodegenerative disorders, appears confusing and not much pertinent to the topic. However, this correlation is important and interesting to understanding the functions of NR4A2 in neuronal development. 

- The authors should better explain the three new cases, which they reported in Table 1, although, as mentioned, the manuscript that reports them is in preparation. These new cases can be a valuable contribution to current research. 

- In this review, the authors found a strong evidence of correlation between NR4A2 variants and neurodevelopmental disorders, such as epilepsy, Psychiatric and Behavioral disorders and language impairment. Further research is needed, especially in the optic of target therapy. The possible treatment in these conditions should be explored.

Author Response

We thank so much the comments of the reviewer.

  1. "Results and discussion" paragraphs could be separated to make the manuscript clearer. In particular, the section, that explore the correlation between NR4A2 and neurodegenerative disorders, appears confusing and not much pertinent to the topic. However, this correlation is important and interesting to understanding the functions of NR4A2 in neuronal development.

A new section on Discussion has been developed, as suggested. We agree that the section on neurodegenerative disorders is important: (1) to review the available and updated data on the importance of specific variants of this gene in the development of Parkinson's disease; and (2) to understand the functions of NR4A2 in neuronal development.

  1. The authors should better explain the three new cases, which they reported in Table 1, although, as mentioned, the manuscript that reports them is in preparation. These new cases can be a valuable contribution to current research.

A new paragraph on this topic has ben incorporated (lines 132 to 151).

  1. In this review, the authors found a strong evidence of correlation between NR4A2 variants and neurodevelopmental disorders, such as epilepsy, Psychiatric and Behavioral disorders and language impairment. Further research is needed, especially in the optic of target therapy. The possible treatment in these conditions should be explored.

Although not specifically developed to treat the NR4A2-related neurodevelopmental disorder, we consider that the different lines of research that are underway to stimulate the activity of NR4A2 protein are highly promising. However, as far as we know, nothing has been published so far in this regard. Probably because it is a very recently known disease and at the moment with few known cases. We hope that the present paper contributes to improving this issue.

Round 2

Reviewer 1 Report

Comments and Suggestions for Authors

No further comments or suggestion.